# An Orthologics Study of the Notch Signaling Pathway

**DOI:** 10.3390/genes15111452

**Published:** 2024-11-10

**Authors:** Wilfred Donald Stein

**Affiliations:** Silberman Institute of Life Sciences, Hebrew University, Jerusalem 91904, Israel; wdstein@mail.huji.ac.il

**Keywords:** evolution, orthologs, notch signaling system, cell-to-cell communication, origin of multicellularity, co-option

## Abstract

The Notch signaling pathway plays a major role in embryological development and in the ongoing life processes of many animals. Its role is to provide cell-to-cell communication in which a Sender cell, bearing membrane-embedded ligands, instructs a Receiver cell, bearing membrane-embedded receptors, to adopt one of two available fates. Elucidating the evolution of this pathway is the topic of this paper, which uses an orthologs approach, providing a comprehensive basis for the study. Using BLAST searches, orthologs were identified for all the 49 components of the Notch signaling pathway. The historical time course of integration of these proteins, as the animals evolved, was elucidated. Insofar as cell-to-cell communication is of relevance only in multicellular animals, it is not surprising that the Notch system became functional only with the evolutionary appearance of Metazoa, the first multicellular animals. Porifera contributed a quarter of the Notch pathway proteins, the Cnidaria brought the total to one-half, but the system reached completion only when humans appeared. A literature search elucidated the roles of the Notch system’s components in modern descendants of the ortholog-contributing ancestors. A single protein, the protein tyrosine kinase (PTK) of the protozoan *Ministeria vibrans*, was identified as a possible pre-Metazoan ancestor of all three of the Notch pathway proteins, DLL, JAG, and NOTCH. A scenario for the evolution of the Notch signaling pathway is presented and described as the co-option of its components, clade by clade, in a repurposing of genes already present in ancestral unicellular organisms.

## 1. Introduction

The Notch signaling pathway is one of the many signaling pathways in biological systems (the collection in https://www.tocris.com/signaling-pathways lists 14 of them, accessed on 5 October 2024).

The Notch pathway plays a major role in embryological development (for an extensive review see [1]), in systemic mastocytos [2], in many cancers [3], in neurological conditions such as autism [4], in osteoporosis [5], and in the ongoing life processes of many animals. In these many roles, one may list the following: neuronal function and development [6], fate determination in angiogenesis [7] and in cardiac development [8], control of the differentiation of both the endocrine and exocrine pancreas [9], cell fate decisions between the secretory and absorptive lineages in the gut [10], and the development of mammary glands [11] and of alveoli in the lungs [12]. 

It is generally accepted that the Notch pathway acts as an arbiter between two alternative cell-fate decisions: thus, it has been said that “Notch acts in a permissive manner: It does not determine the fate of a cell but rather whether a given fate is adopted” (Ehebauer et al. [13]), or, as Newman [14] states, “it enable[s] (by enforcing alternative gene expression states in adjoining cells with a common genome), the regulated coexistence of multiple cell types within a common tissue mass.” As examples of such commitments, one can point to the immune system, where the Notch pathway determines the commitment to development of either the T-cell or the B-cell lineage [15], while in the intestinal crypt, Notch signaling determines whether a cell will follow the secretory or else the absorptive path. “A Notch-activated cell is prevented from changing into a secretory lineage as long as its neighbour cells express sufficient Notch-ligands, a phenomenon known as lateral inhibition” [16]. As additional examples, one might point to how the Notch pathway controls the differentiation of mesenchymal stem cells into adipocytes [17] and point to its key role in kidney development where it establishes the fate of proximal tubular epithelial cells [18].

Figure 1 shows two cells; the one on the left is the cell that we will call the Sender cell, that on the right is the Receiver cell. The Sender cell binds to and interacts with the Receiver cell, controlling its behavior, as discussed above in the Introduction. Embedded in the membrane of the Sender cell are the three DLLs and the two JAG proteins, which are the ligands for the four Notch proteins, also membrane-embedded, of the Receiver cell. On binding of the ligands to the NOTCH proteins, these receptors are hydrolytically cleaved and the intracellular portion of the NOTCH proteins (symbolized in the figure as NICD) move to the cell nucleus (depicted as the circle within the receiver cell) where they interact with the RBP proteins that regulate the HES transcription factors [19]. These transcription factors determine the expression or repression of targeted proteins in the Receiver cell and hence its fate. The central core of the system is thus the DLL/JAG–NOTCH–RBPJ–HES chain, which enables the sender cell to control the expression of proteins in the adjacent receiver cell. Additional members of the pathway activate or repress other components of the chain [20]. An example is the co-repressor complex, composed of six proteins, depicted in Figure 1 as being within the nucleus of the Receiver cell. The proteins of this complex cleave the Notch protein at the inner face of the cell membrane of the Receiver cell. This releases the intracellular domain of the Notch protein (NICD), which then moves to the nucleus, where it can regulate gene expression by activating the RBP transcription factors. Another group of regulating proteins are the six proteins of the cytoplasmic γ-secretase complex, also depicted in the figure. Here, the three DVLs and the two NUMB proteins interact with and repress the NOTCH proteins, as do the five proteins of the DTX family (although these can also activate NOTCH proteins). The extent of these activating or repressing processes are, of course, determined by the epigenetic status of the receiver cell. The Fringe proteins (LFNG, MFNG, and RFNG) are glucosaminyltransferases that have been shown to modulate Notch activity. When they interact with the DLLs, they are strongly activated, but they markedly inhibit signaling when they interact with the JAG ligands. ADAM7 also interacts directly with the NOTCH proteins, regulating their activity. The set of Mastermind-Like proteins, with symbols MAML1, MAML3, and others in this block, are transcriptional co-activators in the Notch signaling pathway [21].

The evolution of such an important system is an intriguing question and has been tackled before. Gazave et al. [19] presented an impressive analysis of the question, but their study comprised less than half of the Notch system proteins depicted in Figure 1. The comprehensive work of Lv et al. [20] is confined to Notch signaling in the invertebrates, while the study of Babonis and Martindale [22], which considers a broad range of signaling systems, deals with only twenty components of the Notch system. The present work considers all 49 of the Notch pathway components (46 that appear in Figure 1 and, in addition, 3 that were identified after the original version of Figure 1 was first built), and considers both vertebrate and invertebrate animals. It is based on the use of orthologs. This provides a firm foundation for elucidating the appearance of the proteins of the Notch signaling system, evolutionary level by evolutionary level, from the first protozoa to modern humans. Indeed, only with the appearance of the first humans does the recruiting to the system reach completion.

The evolution of other signaling pathways has been studied. Examples are the Jak/Stat pathway [23], where a fully active system was already in place in the Bilateria, and the Wnt pathway [24], where the Cnidaria first demonstrated a fully functional system.

## 2. Methods

Searches for orthologs were performed using the protein BLAST (Basic Local Alignment Search Tool) program of the NCBI (National Center for Biotechnology Information of the National Library of Medicine): https://blast.ncbi.nlm.nih.gov/Blast.cgi, accessed on 5 October 2024. The non-redundant protein sequences (nr) database was interrogated: this searches “All non-redundant GenBank CDS translations+PDB+SwissProt+PIR+PRF excluding environmental samples from WGS projects”. Under the heading “Algorithm Parameters”, the following search parameters were always used: Max target sequences: 1000; Expect threshold: 200000; Word size: 2; Max matches in a query range: 0; Matrix: BLOSUM-62; Gap costs existence: 11; Extension: 1; No compositional adjustments; No low-complexity regions filter. BLOSUM-62 was chosen, this being the default matrix for protein BLAST. Experimentation has shown that the BLOSUM-62 matrix is among the best for detecting the weakest protein similarities.

To be recognized as an ortholog in the present study, the appropriate sequence returned by the BLAST search of the databases had to be annotated with the same name as the probe sequence from the human genome. A sequence annotated as “-like” was rejected. Of course, by the definition of an ortholog, the probe sequence had to be absent when earlier-appearing clades were searched. 

Sequence comparisons between two proteins and dot plots of the comparisons were made using the BLAST 2-sequences tool of the BLAST program, using the same algorithm search parameters as above. All one-on-one Expect values recorded in this paper are the results of BLAST 2-sequence comparisons between the two proteins.

Alignments between protein sequences were established and dot plots generated using the COBALT (Constraint-Based Alignment Tool) aligner at the NCBI https://www.ncbi.nlm.nih.gov/tools/cobalt/cobalt.cgi?CMD=Web, accessed on 5 October 2024.

Properties of the orthologs listed in Appendix A were taken from the listings in GeneCards, https://genealacart.genecards.org/Result, accessed on 5 October 2024. GeneCards proved useful also when the aliases of genes discussed in the literature had to be interpreted to provide HGNC symbols.

The clades leading up to the emergence of the primates are numbered as Phylostratum Levels in Table 1, which follows the formulation by Domazet-Loso and Tautz [25]:

## 3. Results

Figure 1 (see the Introduction) depicts 46 proteins of the NOTCH signaling pathway. These are listed in Table 2a,b together with the orthologs found by BLAST searches (see Methods). 

Included in the tables are the three recently found paralogs of the NOTCH2NL gene [26], which appeared in human evolution some three million years ago. Also listed in the tables are the HGNC symbol for each protein (also in cases where the name listed in Figure 1 is not an HGNC symbol), the Uniprot symbol for the protein, the ortholog level, the phylostratum number, and the Expect value obtained by a BLAST 2-sequence comparison between this ortholog and the corresponding protein in *Homo sapiens*. 

Appendix A lists these 49 proteins and the annotations obtained by querying GeneCards (https://www.genecards.org/, accessed on 5 October 2024).

Figure 2 depicts the contribution of the orthologs at each phylostratum level (see the definitions of phylostratum levels in Methods—Table 1), while Figure 3 shows the data as the accumulated total of orthologs as a function of phylostratum level:

Contributions begin already at the level of the protozoa, with major additions occurring with the appearance of the sponges, the sea anemones, and the jawed fish.

Contributions begin already at the level of the protozoa, with major additions occurring with the appearance of the sponges, the sea anemones, and the jawed fish. Interestingly, it would appear that the sea urchins and tunicates did not contribute new proteins to the Notch signaling pathway.

From Figure 3, it would appear that by the time of the arrival of the sea anemones, fully half of the Notch signaling proteins had accumulated to the pathway in its evolutionary development.

Table 3 below shows the proteins that were to form the Notch signaling system as they appeared, clade by clade, with the evolution of each new clade in the early evolutionary history of the animals. It should be appreciated that these proteins accumulate and so, at each clade, these Notch signaling proteins, which appeared in earlier clades, are in general still retained in the newly evolved organism.

Table 3 shows the contribution of orthologs of the proteins of the Notch signaling system for the lower half of the evolutionary trajectory of the animals. Regarding data from Table 2, the genome of the primitive Metazoan, the Placazoa *Trichoplax adherens*, possesses no Notch protein designated as such, but a sequence designated as “uncharacterized protein TRIADDRAFT_57304”; when it is BLASTED against *H. sapiens*, it yields as the two top hits, NOTCH1 and NOTCH2. These show Expect values of 0.0 and 0.0, respectively, in BLAST 2-sequence analyses against this Trichoplax sequence.

The protein that appears as [DLL] among the Porifera components in Table 3 above is designated in this fashion because the five DLL proteins of the sponges do not fit the definition of an ortholog as given in the Methods Section: “To be recognised as an ortholog in the present study, a sequence found when searching the databases had to be annotated with the same name as the probe sequence from the human genome”. However, Figure 1 depicts only three DLL proteins, not five. To accommodate this complexity, the “ortholog” in the sponges is designated here as a single protein [DLL].

The sequences of two of the five DLL proteins of the sponge *Amphimedon Queenslandica* are compared with the sequence of DLL1 of *H. sapiens* in Figure 4 below. 

### Exploring the Deep Evolutionary Origins of the Notch, DLL, and JAG Proteins

Porifera are classified as Metazoa, the earliest multicellular animals. Table 3 shows that both a DLL protein and NOTCH2 had already appeared in these animals, giving them the possibility of cell–cell interactions, the DLL in the Sender cell and the NOTCH in the Receiver cell both being present. It is of interest to ask what might have been the ancestors of these fundamental proteins of the Notch signaling system. To answer this question, we performed a BLAST search, using the NOTCH protein of the glass sponge *Oopsacas minuta* as bait and querying all the animals below Metazoa. The top hit was BAM76481.1, the annotation of a sequence in the protein tyrosine kinase (PTK) of the bacterivorous amoeba *M. vibrans*. A BLAST 2-sequence comparison between this sequence and that of the sponge’s NOTCH returned a convincing Expect value of 2 × 10^−103^. 

A comparison between the protein sequences of the NOTCH protein of the sponge *O. minuta*, the four NOTCH proteins of *H. sapiens*, and the protein tyrosine kinase of *M. vibrans* is shown in Figure 5, as follows:

The glass sponge possesses a single DLL protein named “Delta”. Using “Delta” and querying all the animals below the Metazoa, the same sequence BAM76481.1 of *M. vibrans* was returned. A BLAST 2-sequence comparison between this sequence and that of the sponge’s DLL returned an Expect value of 4 × 10^−80^.

A comparison between the protein sequences of the single Delta protein of the sponge *O. minuta*, three DLL proteins of *H. sapiens,* and the protein tyrosine kinase of *M. vibrans* is shown in Figure 6 that follows:

Finally, the two JAG proteins of the Notch signaling pathway, present also with the DLL proteins on the membrane of the Sender cell (Figure 1), again have strong sequence similarity with the protein tyrosine kinase of the amoeba *M. vibrans*, the BLAST 2 sequence Expect vales being 2 × 10^−120^ for JAG1 and 3 × 10^−122^ for JAG2, respectively. A comparison between the protein sequences of the JAG proteins of *H. sapiens* and the protein tyrosine kinase of *M. vibrans* is shown in Figure 7:

A phylogenetic tree of the relationship between the protein tyrosine kinase (PTK) of *M. vibrans* and the JAG and DLL proteins of *H. sapiens* is shown in Figure 8:

These sequences were further analyzed by dialing down the maximum sequence difference routine of the COBALT’s program to 0.7. After this, only the DLLs and the PTK remained. This showed that the amoeba’s PTK and the DLL sequences were more closely related to each other than either was to the sequences of the JAG proteins. (At a maximum sequence difference of 0.8, all seven proteins still appeared.)

All of the sequences depicted in Figure 5, Figure 6 and Figure 7 show a series of six or more repeating cysteine residues, characteristic of proteins of the EGF (Epidermal Growth Factor) family. This will be considered further in the Discussion section. 

From Table 3 it can be seen that by the time the Cnidaria appeared, the entire central core of the pathway was in place, with two of the membrane-bound ligands of the Sender cell (DLL1 and JAG1) and the nuclear-active terminal protein of the Receiver cell being present, as well as two Notch proteins—the receptors on the Receiver cell. In addition, some proteins serving as activators and repressors of the central core proteins appeared, members of the co-repressor complex (Figure 1) and the γ-secretase complex, as well as the Fringe proteins (LFNG, MFNG, and RFNG). The further evolution of the Notch signaling pathway from the Cephalochordate [27] through to sharks and rays (the Elasmobranchi) and bony fish [28] was of proteins that contribute further to the control of the central core (as listed in Table 2a,b). Notch4, with a role in angiogenesis [7], appeared with the evolution of the amphibia. PTCRA (Pre-T-Cell Antigen Receptor Alpha), which contributes to the regulation of early T-cell development [29], arose with the amniota. The three NOTCH2NL paralogs appeared during the recent evolution of the human species [26].

## 4. Discussion

As noted in the Introduction, the evolution of the Notch signaling system has been tackled before. The valiant studies of Gazave etal [19], Lv etal [20], and Babonis and Martindale [22] did not, however, include all of the 49 components of the system, nor did they cover a comprehensive range of animals from the earliest fungi to modern humans. The present paper fills out those gaps but, in particular, goes beyond those studies. We do not merely note the name of the animal species in which a particular component of the Notch signaling system first appeared, but specify the animal as its ortholog. This ensures (within current knowledge) that the animal in question is the first to have evolved the component in question. In addition, the following section attempts to identify the function of an ortholog in the animal in which it was first found. This emphasizes the important point that an ortholog plays an ascertainable role in the modern descendants of the animal in which it is found—but that role is not as a component of the Notch signaling pathway. The pathway came to be fully operational only with much further evolution.

In the evolutionary trajectory from the first living organisms to the emergence of humans, genes were added to the accumulating genome at each phylostratum level. At many levels, these included genes that were later to form the Notch signaling system. It is of much interest to ask what the function of such a “Notch signaling” gene was in an animal that long preceded the appearance of the complete Notch system.

Table 3 above shows the contribution of orthologs of the proteins of the Notch signaling system at each phylostratum level for the lower half of the evolutionary trajectory that led to humans. At the level of the Eukaryota, we see four genes appearing: RBPJ, SNW1 (denoted as SKIP in Figure 1), HDAC1, and HDAC2. These four are found in the nucleus of the Receiver cell as can be seen in the scheme of Figure 1. The two HDAC genes shown in the figure are in the co-repressor complex while RBPJ is depicted as directly interacting with DNA, and SN1/SKIP interacts with RBPJ.

Of these four genes, HDAC genes are expressed in the cell nucleus of the protozoan *Trypanosoma brucei*, where they participate in the expression of the VSG (Variant Surface Glycoprotein) genes that enable the parasite to evade the host’s immune response [30]. Indeed, the paper [30] reported on attempts to use inhibitors of HDACs to combat infection by the parasite. For RBPJ, Prevorovsky et al. [31] found orthologs of RBPJ in several fungal species, while Oravcova et al. [32] showed that one such expressed protein (named in *Schizosaccharomyces pombe* as Cbf11) could act as a sequence-specific DNA-binding activator of transcription. For SNW1, Halova et al. [33] working with the fungus *Saccharomyces cerevisiae* showed that the splicing factor Prp45 (a homolog of SNW1) was expressed in the yeast. It is present in the spliceosome and takes part in co-transcriptional splicing. 

Thus, all four genes have been shown to be expressed in fungi or protozoa and suggestions have been made regarding their roles in these organisms. Yet, hundreds of millions of years would pass before a complete Sender-to-Receiver Notch signaling system would appear in the Cnidaria (the sea anemones, corals, and jellyfish).

Table 3 shows that, meanwhile, thirteen Notch signaling system orthologs were recruited with the appearance of Porifera (sponges). Schenkelaars et al. [34], noting that sponges were the first animals to be truly multicellular (the earlier fungi forming mere syncitia), investigated whether the cell-to-cell Notch signaling system might already be fully present in a sponge. Investigating the Notch and Wnt signaling pathways in the glass sponge *O. minuta,* they showed that while many components of the Notch pathway were already present in the sponge, including the eponymous NOTCH proteins, key components were not. The nuclear-active HES1 and HES5 of the Receiver cell were not identified nor were JAG1 or JAG2 of the Sender cell. Schenkelaars et al. [34] did find a DLL of the Sender cell, shown as [DLL] in Table 3 above. The DLL in the sponges are not named as either DLL1, DLL3, or DLL4 [see above], although they may play the role in the sponge as a Notch ligand [34]. The extracellular portion of the protein has a strong sequence similarity to that of the human DLL proteins [Figure 4] and it is this portion of these molecules that binds to the NOTCH proteins. Richards and Degnan [35,36], studying the sponge *Amphimedon queenslandica,* followed the expression of Notch pathway components during the embryological development of this organism, with results suggesting that classic roles for the pathway—like mediating the choice between cell fates (here between the path to an anterior pole cell or, alternatively, to a flask cell)—are already operative at this stage of evolution. Richards and Degnan (*loc.cit*) proposed “that the cell-type-restricted expression domains of the *A. queenslandica* Deltas may reflect Notch activity in shaping the development of these particular cell lineages to the exclusion of their neighboring cells”, just as the Notch pathway functions in higher animals. 

Sponges have no nervous system but they do have a communication system, organized around the sponge’s digestive chambers, and involved in the clearing of bacteria or cellular debris [37]. The appearance of a true nervous system had to wait for the Cnidaria, which possess nerves that activate the anthocysts (tentacles) used by these animals to entrap prey. This enabled the adoption of a new lifestyle—active predation as opposed to the more passive filter-feeding of their evolutionary predecessors. The Cnidaria have a full Notch pathway, extending from DLL1 and JAG1, which are ligands in the Sender cell (Figure 1), to the nuclear-active HES1 in the Receiver. Pharmacological inhibition [37] of Notch pathway components (inhibiting presenilin by treatment with the PSEN inhibitor DAPT (N-[N-(3,5-difluorophenacetyl)-L-alanyl]-S-phenylglycine t-butyl ester) or blocking the intracellular domain of NOTCH itself by treatment with the Notch inhibitor SAHM1 (stapled α-Helical peptide derived from MAML) showed in *Hydra* that such treatment disturbed head regeneration and tentacle patterning. These effects demonstrate a functional role for the pathway in the animal’s development [37]. Pan et al. [38] extended these findings, showing that the genetic modification of Notch pathway components (over-expressing the intracellular domain components of NOTCH, or knockdown of NOTCH) recapitulated the pharmacological effects. Note that PSEN1 and PSEN2, inhibited in the experiments of Musser et al. [37], were added to the evolving genome already in Porifera—thus showing that earlier-appearing components of the Notch pathway were integrated into the signaling system by the time that Cnidaria appeared.

One can expect that single-cell genomics studies using perhaps CellChat [39] will add much to the study of cell to cell communication and the role of Notch signaling.

Our exploration of a possible pre-Metazoan origin of the Notch proteins suggests (see Results) a protein tyrosine kinase (PTK) of the amoeboid *M. vibrans* as a possible candidate. To our surprise, our subsequent explorations of such an origin for the DLL and JAG proteins came up with the same protein tyrosine kinase as a possible candidate for these proteins.

The PTK, DLL, JAG, and Notch proteins show extensive sequence similarity (Figure 4, Figure 5, Figure 6 and Figure 7) and, in particular, a forty-membered sequence containing sets of six-fold repeats of Cysteine residues. This is a feature characteristic of the large family of EGF proteins [40], where these sequences form part of the extracellular ligand-binding domain of these proteins. The 42-long sequence GDKCQTDMNECLSEPCKNGGTCSDYVNSYTCKCQAGFDGVHC in NOTCH2 includes the calcium-binding domain of this protein (GeneCards—see Methods).

Suga and his colleagues [41] performed an extensive survey of the PTK proteins of all pre-Metazoan animals. In particular, they analyzed the Filasterea clade to which *M. vibrans* belongs. (“Filasterea is a proposed basal Filozoan clade of single-celled amoeboid eukaryotes that includes Ministeria and Capsaspora.”—Wikipedia). Filasterea are a sister group to Metazoa and Choanoflagellata and are thus central to a discussion of the evolution of Metazoa. Suga et al. [41] concluded that the “divergence pattern of [PTKs] is consistent with the hypothesis that they act to detect changes in the extracellular environment or to recognize and catch prey, because each organism has to be adapted to its own environment and nutrient conditions”, a statement that provides a plausible answer to the question of what the role of the PTKs was in these pre-Metazoan animals.

The Filasterean PTKs have transmembrane sequences, as do other members of the EGF protein family. It has not been shown that in these pre-Metazoan PTKs, dimerization of the transmembrane portions of the molecule plays an important role in their function as tyrosine kinases. This has been found, however, in Metazoan PTKs [42].

But it was from their extracellular sequences that the great advance came. Instead of these extracellular receptor sequences recognizing and binding to nutrients and other components of the environment, they evolved the ability to recognize and bind to each other—and also evolved new non-coding sequences that caused them to be expressed in two different, but contiguous, cells.

It is thus a possible scenario that, as Metazoans diverged from their common Filasterean/Metazoan precursor, their protein tyrosine kinases evolved into forms that would recognize and bind to one another in the typical ligand/receptor mode. Evolving from a pre-Metazoan protein tyrosine kinase, the DLL and JAG proteins became ligands on the extracellular surface of Sender cells while the Notch proteins became the extracellular receptors on Receiver cells. Thus, the two cells could bind to one another and send and receive developmental information commands.

## 5. Conclusions

This paper, using BLAST searches, identified orthologs for all of the 49 components of the Notch signaling pathway, and elucidated the historical time course of the integration of these proteins, as the relevant animals evolved. Cell-to-cell communication is an inescapable necessity for the functioning of multicellular animals. The Notch signaling system is a prime example of such an actor in cell-to-cell communication, with, as we have described, its Sender cell communicating instructions to the Receiver cell that determine the Receiver’s fate. It is not surprising, therefore, that the Notch system reached a fully working configuration with the evolutionary appearance of Metazoa, the first multicellular animals (as demonstrated in the description above of experiments on the glass sponge [35,36]). Yet, this configuration, as the present paper documents, includes many proteins whose appearance in evolution long preceded the appearance of Metazoa. At every evolutionary level at which these components first appeared in an animal, they play a discernible role in the life processes of the contemporary descendants of that ancestral animal. But these roles are not concerned with cell-to-cell communication, and could not have been so until multicellular animals themselves appeared. Such a linking together of pre-existing components into a workable system is a typical case of the co-option that is so often found in evolution, “the repurposing of genes [that are] present in ancestral unicellular organisms owing to the changed scale of the multicellular state”, as Newman describes it in his cogent discussion of the major transitions in animal evolution [14]. The patterns of co-option reflect a complex interaction between the retention of ancestral forms of a gene and its evolutionarily conserved mutations, “combined with changes in function (what a gene does), in time (when a gene is activated), and in space (where a gene is activated)” [43]. Indeed, as a further example, a whole developmental scenario can be co-opted, as Litman and Stein discussed [44], for how skin placodes were co-opted in the evolution of teeth, hair, and eccrine glands.

## Figures and Tables

**Figure 1 genes-15-01452-f001:**
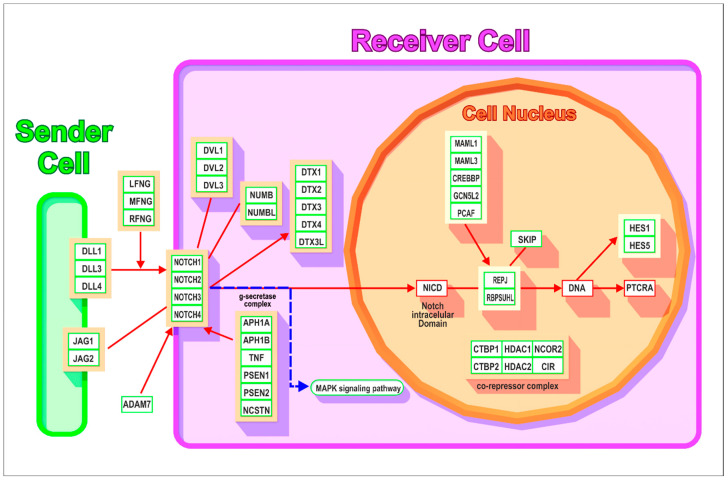
Redrawn from WSP268, the version of the Notch signaling pathway depicted on Wikipedia https://www.wikipathways.org/pathways/WP268.html, accessed on 5 October 2024. On the left, in green, is the Sender cell. On the right, in red, is the Receiver cell. The Nucleus is depicted within the Receiver cell. The gene names and symbols are depicted in blocks on the figure, discussed directly below and listed, with additional information, in Table 2a,b and Appendix A.

**Figure 2 genes-15-01452-f002:**
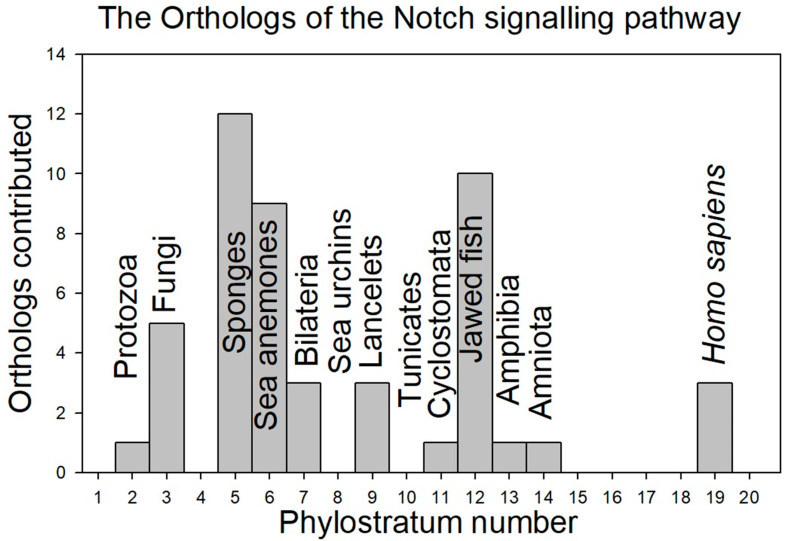
The contribution of orthologs of the proteins of the Notch signaling system, phylostratum level by phylostratum level. Column 4 in each of Table 2a,b, which lists the phylostratum number for each ortholog identified, provides the data on which this figure was built.

**Figure 3 genes-15-01452-f003:**
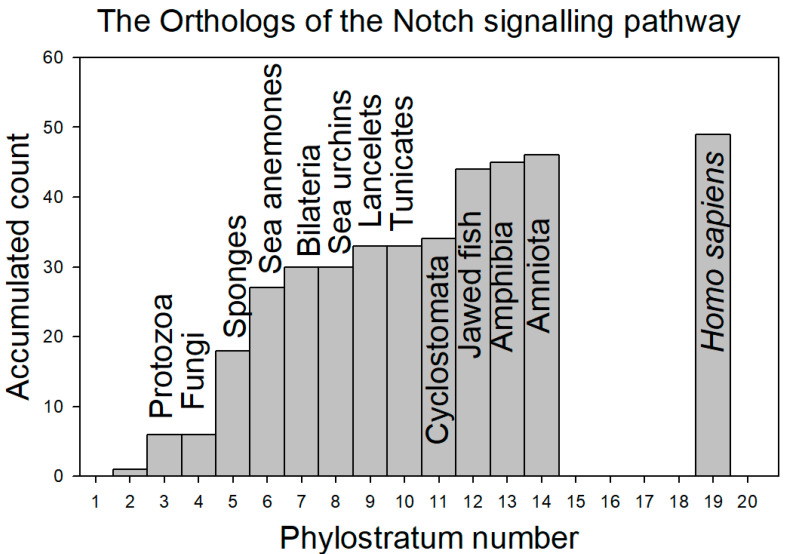
The accumulated contribution of orthologs of the proteins of the Notch signaling system per phylostratum level. Column 4 in each of Table 2a,b, which lists the phylostratum number for each ortholog identified, provides the data on which this figure was built. It would appear that by the time of the arrival of the sea anemones, fully half of the Notch signaling proteins had accumulated to the pathway in its evolutionary development.

**Figure 4 genes-15-01452-f004:**
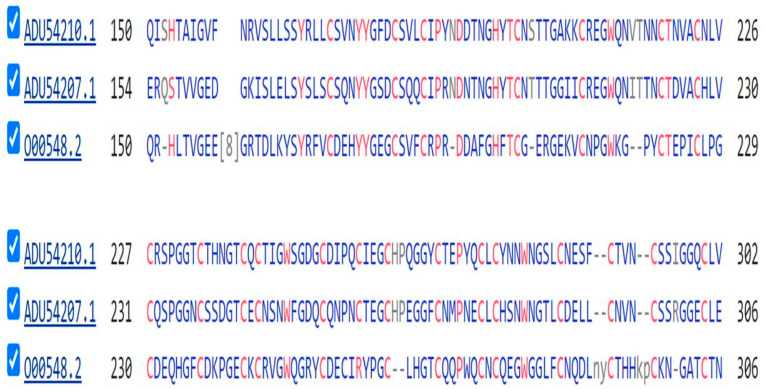
A comparison of a portion of the protein sequences of three DLL proteins. The top two sequences are for Delta5 and Delta2 of *Amphimedon Queenslandica* while the lowest is that of DLL1 of *H. sapiens*. The repeated Cysteine Residues (Cs) are, in most cases, shared by all three proteins.

**Figure 5 genes-15-01452-f005:**
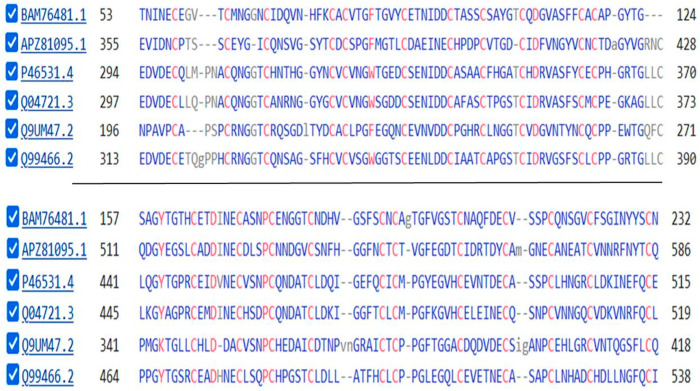
A comparison of a portion of the protein sequences of the protein tyrosine kinase of *M. vibrans*, the NOTCH protein of *O. minuta*, and the four NOTCH proteins of *H. sapiens* proteins. The top sequence in each group is that of the protein tyrosine kinase and the next is that of the sponge, while the next four sequences are for the NOTCH proteins of *H. sapiens*. The repeated Cysteine Residues (Cs) are, in most cases, shared by all three proteins and there are additional residues that are shared across the proteins.

**Figure 6 genes-15-01452-f006:**
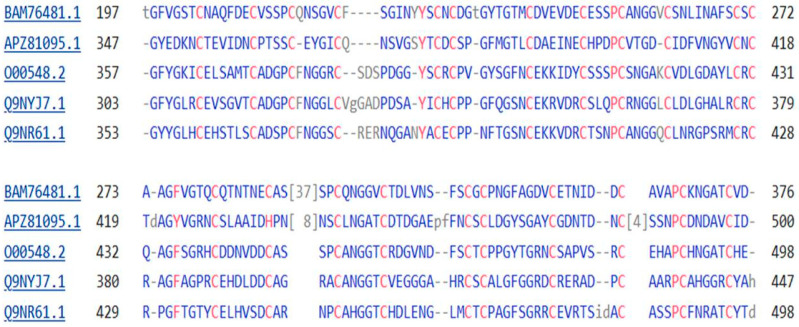
A comparison of a portion of the protein sequences of the protein tyrosine kinase of *M. vibrans*, the Delta protein of the sponge *O. minuta*, and the protein sequences of three DLL proteins of *H. sapiens*. The top sequence in each group is that of the protein tyrosine kinase and the next is that of the sponge Delta protein, while the next three sequences are for the DLL proteins of *H. sapiens*. The repeated Cysteine Residues (Cs) are, in most cases, shared by all three proteins and there are additional residues that are shared across the proteins.

**Figure 7 genes-15-01452-f007:**
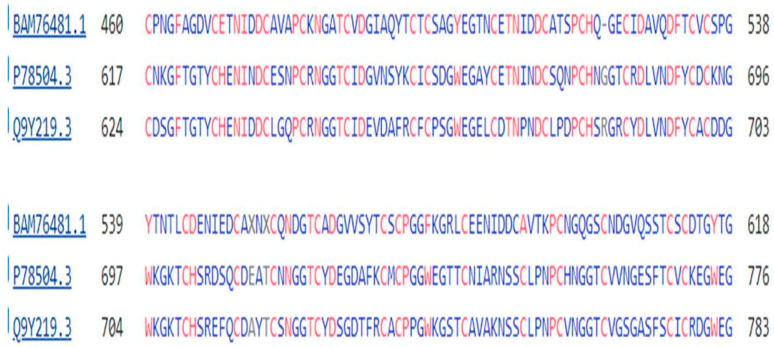
A comparison of a portion of the protein sequences of the protein tyrosine kinase of *M. vibrans*, and the JAG proteins of *H. sapiens*. The top sequence in each group is that of the protein tyrosine kinase, while the next two sequences are for JAG1 and JAG2 of *H. sapiens*. The repeated Cysteine Residues (Cs) are, in most cases, shared by all three proteins and there are additional residues that are shared across the proteins.

**Figure 8 genes-15-01452-f008:**
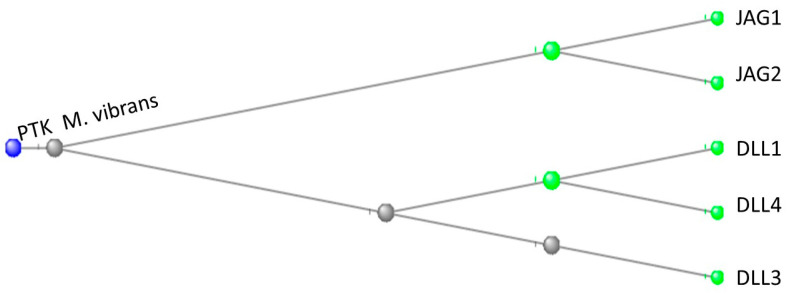
A phylogenetic tree depicting the relationship between the PTK of *M. vibrans* and the JAG and DLL proteins of *H. sapiens*. The phylogenetic tree is rooted at the PTK of *M. vibrans*. The JAG proteins cluster together as do the DLLs.

**Table 1 genes-15-01452-t001:** Definition of the 19 phylostrata.

Phylostratum Number	Clade Beginning to Onset of Next	Examples of Phylostratum Members
1	All life up to Eukaryota	Eubacteria and bacteria
2	Eukaryota to Opisthokonta	Unicellular nucleated cells
3	Opisthokonta to Holozoa	Yeasts and molds
4	Holozoa to Metazoa	Choanoflagellates
5	Metazoa to Eumetazoa	Sponges and jellyfish
6	Eumetazoa to Bilateria	Sea anemones
7	Bilateria to Deuterostomia	Worms, limpets and octopus
8	Deuterostomia to Chordata	Sea urchins
9	Chordata to Olfactores	Lancelets
10	Olfactores to Craniata	Sea squirts
11	Craniata to Euteleostomi	Lampreys
12	Euteleostomi to Tetrapoda	Jawed fish
13	Tetrapoda to Amniota	Frogs and toads
14	Amniota to Mammalia	Birds and reptiles
15	Mammalia to Eutheria	Platypus (15.1); Opossum (15.2)
16	Eutheria to Boreotheria	Early placental animals (elephant, armadillo)
17	Boreotheria to Euarchontoglires	Hoofed and pawed animals
18	Euarchontaglires to Primata	Rabbits and rodents
19	Primata	Monkeys (19.1); great apes (19.2)

**Table 2 genes-15-01452-t002:** (**a**) Orthologs of the notch signaling genes (A to H). (**b**) Orthologs of the notch signaling genes (J to T).

**(a)**				
**HGNC (or Name on WP268)**	**Uniprot**	**Ortholog Found in the Following:**	**PSn ***	**E Value ****
ADAM17	P78536	Branchiostomatidae	9	6.00E-178
APH1A	Q96BI3	Fungi	3	1.00E-34
APH1B	Q8WW43	Fungi	3	3.00E-40
CIR1 (CIR)	Q86X95	Porifera	5	1.00E-63
CREBBP	Q92793	Porifera	5	0.00E+00
CTBP1	Q13363	Bilateria	7	0.00E+00
CTBP2	P56545	Branchiostomatidae	9	0.00E+00
DLL1	O00548	Cnidaria	6	2.0010^-110^
DLL3	Q9NYJ7	Elasmobranchi sharks and rays	12	1.0010^-111^
DLL4	Q9NR61	Branchiostomatidae	9	3.0010^-180^
DTX1	Q86Y01	Bilateria	7	5.0010^-100^
DTX2	Q86UW9	Porifera	5	6.0010^-83^
DTX3	Q8N9I9	Porifera	5	8.001010^-63^
DTX3L	Q8TDB6	Elasmobranchi sharks and rays	12	9.001010^-59^
DTX4	Q9Y2E6	Cnidaria	6	2.001010^-87^
DVL1	O14640	Elasmobranchi sharks and rays	12	0.001010^+00^
DVL2	O14641	Elasmobranchi sharks and rays	12	0.001010^+00^
DVL3	Q92997	Cyclostomata	11	0.001010^+00^
KAT2A (GCN5L2)	Q92830	Porifera	5	0.001010^+00^
HDAC1	Q13547	Fungi	3	0.001010^+00^
HDAC2	Q92769	Protista	2	0.001010^+00^
HES1	Q14469	Cnidaria	6	6.001010^-50^
HES5	Q5TA89	Elasmobranchi sharks and rays	12	5.001010^-43^
**(b)**				
**HGNC (and Name on WP268)**	**Uniprot**	**Ortholog Found in the Following:**	**PSn ***	**10 Valu10 ****
JAG1	P78504	Cnidaria	6	0.0010^+00^
JAG2	Q9Y219	Elasmobranchi sharks and rays	12	0.0010^+00^
LFNG	Q8NES3	Porifera	5	8.0010^-51^
MAML1	Q92585	Elasmobranchi sharks and rays	12	0.0010^+00^
MAML3	Q96JK9	Elasmobranchi sharks and rays	12	0.0010^+00^
MFNG	O00587	Cnidaria	6	3.0010^-91^
NCOR2	Q9Y618	Cnidaria	6	2.2010^-117^
NCSTN	Q92542	Porifera	5	1.0010^-106^
NOTCH1	P46531	Cnidaria	6	0.0010^+00^
NOTCH2	Q04721	Porifera	5	0.0010^+00^
NOTCH2NLA	Q7Z3S9	Humans	19	Idntical
NOTCH2NLB	P0DPK3	Humans	19	Idntical
NOTCH2NLC	P0DPK4	Humans	19	Idntical
NOTCH3	Q9UM47	Elasmobranchi sharks and rays	12	0.0010^+00^
NOTCH4	Q99466	Amphibia	13	0.0010^+00^
NUMB	P49757	Porifera	5	2.0010^-47^
NUMBL	Q9Y6R0	Bilateria	7	4.0010^-111^
KAT2B (PCAF)	Q92831	Cnidaria	6	0
PSEN1	P49768	Porifera	5	210^-132^ as PSN
PSEN2	P49810	Porifera	5	310^-122^
PTCRA	Q6ISU1	Amniota	14	1.0010^-34^
RBPJ	Q06330	Fungus	3	9.0010^-53^
RBPJL (RBPSUHL)	Q9UBG7	Porifera	5	2.0010^-150^
RFNG	Q9Y644	Cnidaria	6	7.0010^-88^
SNW1 (SKIP)	Q13573	Fungus	3	210^-170^
TNF	P01375	Elasmobranchi sharks and rays	12	5.0010^-34^

* Phylostratum number. ** Expect value in BLAST 2-sequence comparison. Appendix A lists these 49 proteins and annotations obtained by querying GeneCards (https://www.genecards.org/, accessed on 5 October 2024).

**Table 3 genes-15-01452-t003:** Contribution of orthologs clade by clade—in earliest clades *.

CLADE	Sender Cell Membrane	FNG Family	Notch Family	γ-Secretase Complex		Co-Repressor Complex	Transcription Factors
Eumetazoa (Cnidaria)	DLL1 JAG1	RFNG MFNG	NOTCH1		DTX4	PCAF NCOR2 CREBBP	HES1
Metazoa (Porifera)	[DLL]	LFNG	NOTCH2	NUMB PSEN1 PSEN2 NCSTN	DTX2 DTX3	GCN5L2 RBPSUHL CIR	
Eukaryota (Fungi)				APH1A APH1B		RBPJ HDAC1 HDAC2 SNW1	

* Data taken from Table 2a,b.

## Data Availability

Not applicable.

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
