# Peer review of "An Orthologics Study of the Notch Signaling Pathway"

_genes, 2024, doi:10.3390/genes15111452_

Round 1
Reviewer 1 Report
Comments and Suggestions for Authors
In this study, the orthologs were identified for all of the 49 listed components of the Notch signaling pathway. The historical time course of the integration of these proteins as the evolution of the animals provided was elucidated. However, some concerns should be given to show the significance of this study.
1. In Figure 1, Copy of WSP268, the version of the Notch signaling pathway depicted in Wikipedia https://www.wikipathways.org/pathways/WP268.html. In this study, the figure from Wikipedia may not be suitable for study. The figure may be re-drawn and analyzed in detail.
2. In Figure 2, The clades leading to Homo sapiens. The numbering of the 19 Phylostratum Levels follows that of [12]. What is the meaning of this figure? It is a unclear figure for publication.
3. In Figure 5, Contribution of orthologs of the proteins of the Notch signaling system for the lower half of the evolutionary trajectory of the animals. This figure is not suitable for publication. What is the meaning of this figure?
4. Figure 10 and 11 may be combined. Figure 11 should be revised and it is a unclear meaning figure.
5. The part of discussion should be revised in detail. The title is An orthologics study of the Notch signalling pathway. The part of discussion should be revised according to the topic of this title.
Comments on the Quality of English LanguageEnglish should be revised by a native English.
Author Response
Reply to REVIEWER 1
I thank the reviewer for the helpful comments and suggestions that have definitely improved the manuscript.
(My replies are in red in what follows)
Top of Form
Comments and Suggestions for Authors
The manuscript explores the evolution of the Notch signaling pathway using orthologs to trace the appearance of 49 key components in various clades, from early protozoans to humans. The study uses bioinformatic tools such as BLAST to identify orthologs across different species and discusses the functional roles of these proteins. This is a well-structured paper that tackles an interesting evolutionary question, but the manuscript could benefit from more recent references, expanded explanations in the results and discussion sections, and minor grammatical improvements.
Additional references were added - see below, and some improvements were made in the grammar.
Abstract:
Line 7: The phrase "to build a firm base for such a study" is somewhat vague. Consider rephrasing to "to provide a comprehensive basis for this evolutionary analysis."
Suggestion taken up
Line 12-15: The sentence "the Notch system reached a fully-working configuration only with the evolutionary appearance of the Metazoa" should specify what constitutes a "fully-working configuration." Clarify how the Metazoan system differs from earlier iterations.
Suggestion taken up
Recommendation: Consider including quantitative results (e.g., the number of orthologs identified at each phylostratum level) in the abstract to make it more data-driven.
Suggestion taken up
Introduction:
Line 33-38: The introductory paragraph about the role of Notch signaling could benefit from additional recent references, particularly regarding its role in human diseases like cancer or neurodevelopmental disorders.
Suggestion taken up
Line 44-50: The explanation of Notch signaling as a binary fate-determining system is clear, but it would be helpful to provide more context for how this function has evolved.
Suggestion taken up and several references discussed and added
Mention specific animal systems or developmental processes in which the Notch pathway has been extensively studied.
Suggestion taken up
The paper should mention Notch pathway as some disease therapeutic targets, such as for osteoporosis reported in Icariin improves osteoporosis, inhibits the expression of PPARgamma, C/EBPalpha, FABP4 mRNA, N1ICD and jagged1 proteins, and increases Notch2 mRNA in ovariectomized rats,2017.
I felt this was unnecessary, after the additions that I had made and did not take up this suggestion.
And kidney disease reported by Notch in the kidney: development and disease,2012
Suggestion taken up
Methods:
Line 57-63: The parameters used for the BLAST search (e.g., Expect threshold, matrix) are clearly outlined, but consider providing more justification for these choices. For example, why was the BLOSUM62 matrix selected over others?
Suggestion taken up
Line 70-75: The use of the phylostratum level system is appropriate, but the text could explain the significance of each phylostratum in more detail. Add a brief summary of what each level represents in evolutionary terms.
Suggestion taken up and a new Table (Table 1) added defining the phylostratum levels in detail.
Results:
Table 1a and 1b: The tables listing the orthologs are informative, but ensure that all abbreviations (e.g., PSn, E value) are clearly defined in the table legend for clarity.
The referee seems to have missed the footnote where these abbreviations are in fact listed. The position of the footnote has been altered to make it clearer.
Line 180-183: The discovery of DLL proteins in sponges is notable, but it would be helpful to expand on the significance of this finding. How do these early DLL proteins compare functionally to their counterparts in higher organisms?
Suggestion taken up and the discussion expanded.
Figure 3: This figure depicting the contribution of orthologs per phylostratum level is informative, but the legend should include a brief explanation of the methodology used to accumulate these contributions.
Suggestion taken up
Line 220-225: The comparison between the PTK protein of Ministeria vibrans and Notch-related proteins is an interesting finding. Expand on the implications of this result. How does this discovery advance our understanding of the evolutionary origin of Notch signaling?
Suggestion taken up and the discussion here expanded
Discussion:
Line 280-285: The discussion of the co-option of proteins is intriguing, but the manuscript would benefit from more specific examples of other signaling pathways where this process has been observed. Provide more context or references to similar evolutionary studies.
Suggestion taken up and the discussion here expanded with additional references added
Line 290-300: When discussing the recruitment of orthologs over time, it would be helpful to include a comparison between the Notch system and other cell-cell communication systems that may have evolved concurrently.
Suggestion taken up and the discussion here expanded
cell-cell communication is the core of this paper and the methodology of investigation of cell-cell communication should be mentioned, recent single cell studies such as Genetic Analysis Uncovers Potential Mechanisms Linking Juvenile ldiopathic Arthritisto Breast Cancer: A BioinformaticPilot Study,2024 and Identification of the novel exhausted T cell CD8 + markers in breast cancer,2024 should be mentioned as example for the application of the single-cell analysis, cell chat (Inference and analysis of cell-cell communication using CellChat,2021) should also been mentioned for single cell analysis to study inter cell communication.
All these suggestions were taken up.
Line 325-330: The mention of Cnidarian species having a nearly complete Notch system could be expanded. What evolutionary pressures or developmental processes might have driven the emergence of this system at this stage in evolution?
This very helpful suggestion was taken up and the discussion here expanded
Submission Date
06 October 2024
Date of this review
15 Oct 2024 12:27:11
Bottom of Form
© 1996-2024 MDPI (Basel, Switzerland) unless otherwise stated

Reviewer 2 Report
Comments and Suggestions for Authors
The manuscript explores the evolution of the Notch signaling pathway using orthologs to trace the appearance of 49 key components in various clades, from early protozoans to humans. The study uses bioinformatic tools such as BLAST to identify orthologs across different species and discusses the functional roles of these proteins. This is a well-structured paper that tackles an interesting evolutionary question, but the manuscript could benefit from more recent references, expanded explanations in the results and discussion sections, and minor grammatical improvements.
Abstract:
Line 7: The phrase "to build a firm base for such a study" is somewhat vague. Consider rephrasing to "to provide a comprehensive basis for this evolutionary analysis."
Line 12-15: The sentence "the Notch system reached a fully-working configuration only with the evolutionary appearance of the Metazoa" should specify what constitutes a "fully-working configuration." Clarify how the Metazoan system differs from earlier iterations.
Recommendation: Consider including quantitative results (e.g., the number of orthologs identified at each phylostratum level) in the abstract to make it more data-driven.
Introduction:
Line 33-38: The introductory paragraph about the role of Notch signaling could benefit from additional recent references, particularly regarding its role in human diseases like cancer or neurodevelopmental disorders.
Line 44-50: The explanation of Notch signaling as a binary fate-determining system is clear, but it would be helpful to provide more context for how this function has evolved. Mention specific animal systems or developmental processes in which the Notch pathway has been extensively studied. The paper should mention Notch pathway as some disease therapeutic targets, such as for osteoporosis reported in Icariin improves osteoporosis, inhibits the expression of PPARgamma, C/EBPalpha, FABP4 mRNA, N1ICD and jagged1 proteins, and increases Notch2 mRNA in ovariectomized rats,2017. And kidney disease reported by Notch in the kidney: development and disease,2012
Methods:
Line 57-63: The parameters used for the BLAST search (e.g., Expect threshold, matrix) are clearly outlined, but consider providing more justification for these choices. For example, why was the BLOSUM62 matrix selected over others?
Line 70-75: The use of the phylostratum level system is appropriate, but the text could explain the significance of each phylostratum in more detail. Add a brief summary of what each level represents in evolutionary terms.
Results:
Table 1a and 1b: The tables listing the orthologs are informative, but ensure that all abbreviations (e.g., PSn, E value) are clearly defined in the table legend for clarity.
Line 180-183: The discovery of DLL proteins in sponges is notable, but it would be helpful to expand on the significance of this finding. How do these early DLL proteins compare functionally to their counterparts in higher organisms?
Figure 3: This figure depicting the contribution of orthologs per phylostratum level is informative, but the legend should include a brief explanation of the methodology used to accumulate these contributions.
Line 220-225: The comparison between the PTK protein of Ministeria vibrans and Notch-related proteins is an interesting finding. Expand on the implications of this result. How does this discovery advance our understanding of the evolutionary origin of Notch signaling?
Discussion:
Line 280-285: The discussion of the co-option of proteins is intriguing, but the manuscript would benefit from more specific examples of other signaling pathways where this process has been observed. Provide more context or references to similar evolutionary studies.
Line 290-300: When discussing the recruitment of orthologs over time, it would be helpful to include a comparison between the Notch system and other cell-cell communication systems that may have evolved concurrently.cell-cell communication is the core of this paper and the methodology of investigation of cell-cell communication should be mentioned, recent single cell studies such as Genetic Analysis Uncovers Potential Mechanisms Linking Juvenile ldiopathic Arthritisto Breast Cancer: A BioinformaticPilot Study,2024 and Identification of the novel exhausted T cell CD8 + markers in breast cancer,2024 should be mentioned as example for the application of the single-cell analysis, cell chat (Inference and analysis of cell-cell communication using CellChat,2021) should also been mentioned for single cell analysis to study inter cell communication.
Line 325-330: The mention of Cnidarian species having a nearly complete Notch system could be expanded. What evolutionary pressures or developmental processes might have driven the emergence of this system at this stage in evolution?
Author Response
REPLY TO REVIEWER 2
I thank the reviewer for the helpful comments and suggestions that have definitely improved the manuscript.
(My replies in red in what follows)
Comments and Suggestions for Authors
In this study, the orthologs were identified for all of the 49 listed components of the Notch signaling pathway. The historical time course of the integration of these proteins as the evolution of the animals provided was elucidated. However, some concerns should be given to show the significance of this study.
- In Figure 1, Copy of WSP268, the version of the Notch signaling pathway depicted in Wikipedia https://www.wikipathways.org/pathways/WP268.html. In this study, the figure from Wikipedia may not be suitable for study. The figure may be re-drawn and analyzed in detail.
Suggestion taken up and a new figure made and substituted for the original and is analysed in more detail in the text
- In Figure 2, The clades leading to Homo sapiens. The numbering of the 19 Phylostratum Levels follows that of [12]. What is the meaning of this figure? It is a unclear figure for publication.
Suggestion taken up and a new Table made (Table 1) and substituted for the original figure – the Table expands on the information in the figure and defines the phylostratum levels in detail.
- In Figure 5, Contribution of orthologs of the proteins of the Notch signaling system for the lower half of the evolutionary trajectory of the animals. This figure is not suitable for publication. What is the meaning of this figure?
Suggestion taken up and a new figure made and substituted for the original and analysed in more detail
- Figure 10 and 11 may be combined. Figure 11 should be revised and it is a unclear meaning figure.
Figure 11 was removed and the information that was in it, put into the text.
- The part of discussion should be revised in detail. The title is An orthologics study of the Notch signalling pathway. The part of discussion should be revised according to the topic of this title.
Suggestion taken up and more material added to the text so as to strengthen the discussion
Comments on the Quality of English Language
English should be revised by a native English.
I cannot accept this suggestion at all. I am a native English speaker, the son of native English speakers, all of us growing up and going to school in English-speaking countries. Indeed, I received the prize in English in my final year at my English-speaking high school. I have written four well-accepted textbooks and over 300 peer-reviewed papers and in all this have never received any unfavourable comment on my English. Rather, I have received compliments on the clarity of my writing.
Nevertheless, the manuscript that I now submit has been reviewed and agreed upon by a native English speaker
The reviewer may perhaps have been worried by the phrase “Cell-to-cell communication is the ineluctable modality of multicellular animals.”. This is, of course, correct English being used by James Joyce in Ulysses: “the ineluctable modality of the visible” - and similar phrases. It expresses perfectly what I meant to convey. But to remove the confusion that this phrase might incur in readers unfamiliar with English literature I have omitted it and replaced it by simpler words.

Round 2
Reviewer 1 Report
Comments and Suggestions for Authors
Some revisions have been made on this manuscript. However, some figures are not clear enough for publication. Please revise according to the requirement of this journal.
Comments on the Quality of English LanguageSome minor revisions should be made on English. Please check it carefully.
Author Response

(The authors gave the same response as above.)

Reviewer 2 Report
Comments and Suggestions for Authors
The manuscript presents an insightful study into the Notch signaling pathway's evolutionary progression, exploring orthologs across various species. While the topic is compelling, certain areas require refinement for improved clarity and scientific rigor. Below are key suggestions for enhancing the manuscript
The manuscript has multiple grammatical errors that affect readability. For instance, in the abstract, “providing a firm basecomprehensive basis for such athe study” should be corrected to “providing a comprehensive basis for the study.” In the introduction, phrases such as “many of these cases” could be more specific and rephrased for clarity. Notch signaling pathway function in many human disease, this should be mentioned to provide background for reader, such as bone cells reported in “Icariin improves osteoporosis, inhibits the expression of PPARgamma, C/EBPalpha, FABP4 mRNA, N1ICD and jagged1 proteins, and increases Notch2 mRNA in ovariectomized rats, 2017” and Systemic mastocytosis reported in Aggressive systemic mastocytosis with the co-occurrence of PRKG2::PDGFRB, KAT6A::NCOA2, and RXRA::NOTCH1 fusion transcripts and a heterozygous RUNX1 frameshift mutation, 2024. Various sentences are long and complex, which can make comprehension challenging. For example, the sentence on page 12 could be revised for clarity and conciseness.
There is a need for additional references in sections discussing broad claims about evolutionary processes. Specifically, statements on the role of Notch in vertebrate evolution and early multicellular organisms would benefit from further supporting citations to ensure accuracy and provide context for readers less familiar with the field.
Consider restructuring parts of the discussion to distinguish findings from previous studies. This would make it easier for readers to understand what this study adds to the existing literature. The methods section could benefit from additional detail, particularly on BLAST search criteria, to ensure reproducibility.
The legends for figures should be more detailed, allowing them to stand alone without extensive reference to the main text. Figure descriptions such as those for Figures 5 and 9 could be expanded to include concise summaries of the main findings they illustrate.
With revisions to address grammar, additional references, and structural improvements, this manuscript could make a valuable contribution to the evolutionary study of the Notch signaling pathway.
Author Response

(The authors gave the same response as above.)
